# Positive Emotion Amplification by Representing Excitement Scene with TV Chat Agents

**DOI:** 10.3390/s20247330

**Published:** 2020-12-21

**Authors:** Shogo Nishimura, Daiki Kimata, Wataru Sato, Masayuki Kanbara, Yuichiro Fujimoto, Hirokazu Kato, Norihiro Hagita

**Affiliations:** 1Graduate School of Science and Technology, Nara Institute of Science and Technology, Ikoma, Nara 630-0192, Japan; kimata.daiki.ju0@is.naist.jp (D.K.); kanbara@is.naist.jp (M.K.); yfujimoto@is.naist.jp (Y.F.); kato@is.naist.jp (H.K.); 2Psychological Process Team, Robotics Project, BZP, RIKEN, Kyoto 619-0288, Japan; wataru.sato.ya@riken.jp; 3Art Science Department, Osaka University of Arts, Osaka 585-8555, Japan; hagita@atr.jp

**Keywords:** human-robot interaction, TV chat agent, emotion amplification, motivation stimulation

## Abstract

This paper proposes emotion amplification for TV chat agents allowing users to get more excited in TV sports programs, and a model that estimates the excitement level of TV programs based on the number of social comment posts. The proposed model extracts the exciting intervals from social comments to the program scenes. By synchronizing recorded video streams and the intervals, the agents may talk with the user dynamically changing the frequency and volume of upbeat utterances, increasing the excitement of the user. To test these agents, participants watched TV content under three conditions: without an agent, with four agents that utter with a flat voice, and with four agents with emotion amplification. Results from 24 young adult Japanese individuals showed that their arousal of participants’ subjective and physiological emotional responses were boosted because of the agents, enhancing their motivation to interact with the agent in the future. With empirical evidence, this paper supports these expectations and demonstrates that these agents can amplify the positive emotions of TV watchers, enhancing their motivation to interact with the agent in the future.

## 1. Introduction

As the number of single households in Japan increases in all age demographics, driven by the increasing unmarried rate and the tendency to form nuclear family units, a lack of communication has become a serious social problem. The Ministry of Internal Affairs and Communications [1] predicts that there will be ∼40% single-person households in 2040. In Japan, a Cabinet Office survey [2] for elderly people living alone reported that 7.0% of the respondents had “almost no conversation" with other people, contrasting with those households of two or more people (2.2%) and single households in other countries (USA, 1.6%; Germany, 3.7%; Sweden, 1.7%). Moreover, the lack of communication is not limited to elders. More than 60% of young people consider the Internet as their place of residence [3], and among them, the occupancy rate of the Internet exceeds those of schools, workplaces, and other areas. Although most of young people in survey communicated via the Internet, they felt dissatisfied or anxious about the quality of their communications. Thus, according to young people, good communication is more difficult in a single-household environment. The decrease in communication opportunities are the likely cause of decreased physical function, deteriorated living environment, depression and dementia. Therefore, activating daily communication is an important aspect in human health. Daily communication activation by agents, such as dialog robots, has been proposed for users lacking natural communication [4,5,6]. Kanda et al. [7] stated that humans and robots need to build a friendly relationship. If an agent, represented by a dialog robot, can build a trusting relationship with a user, it can encourage behavioral changes and provide independent support to the user. Current developments of user-interactive agents have focused on behaviors that elicit amusement, gestures, and speech synthesis [8,9,10]. However, understanding the context of a user’s utterance and providing a flexible reply are difficult tasks for agents because their dialog is typically controlled on a per-scenario scenario basis. Moreover, monotonous utterance content and lack of interest are thought to demotivate the continued use of a dialog agent.

To overcome the existing limitations in communication agents, researchers have focused on the continuous dialog to maintain the user’s desire to use the agent. Here, using TV chat agents, we propose a lively mood presentation system.

## 2. Related Work

### 2.1. Dialog Agents

Dialog agents employ one of two categories of dialog systems (task-oriented dialog systems, such as traffic guidance and seat reservation that perform purpose-driven utterances, and non-task-oriented dialog systems, in which the dialog itself is the objective). In non-task-oriented dialogs, such as chat, the dialog must be continued by motivating the user’s willingness to interact with the agent. Many studies have attempted to continue the dialog by semantically connecting the utterances. For example, Tsunomori et al. [11] proposed a system that analyzes human dialog and web data, prepares a large-scale dialog database, and outputs the sentences judged as the most natural responses to user input.

The dialog between the user and the agent is not sustained by semantic consistency alone. If the robot’s utterances maintain semantic consistency, users may get tired, and thus, long-term interaction will eventually fail. Therefore, the agent must arouse the motivation of the user to talk with it. Miyazawa et al. [12] presented information with unexpectedness and novelty, and introduced humanity and other social aspects into the dialog. These high-level social skills are expected to promote continued use of the dialog system. They highlighted the importance of introducing the two novel features in their system.

To improve the sociality and novelty of interactive agents, Minami et al. [13] developed a TV chat agent system that uses comments on social networking services (SNS), which designed for chatting while watching TV programs. The agent reads out the SNS comments and repeats the user’s utterances. By reading the SNS comments, the agent speaks in a context that matches the content of the TV program. However, behaviors such as appropriate utterance intervals of agents were not discussed.

Iio et al. [14] considered the sociality of agents and investigated the effect of multiple dialog robots on the user’s sense of dialog. After talking with one, two, or three robots, the users evaluated their impression of the robot. A conversation with three robots was considered as “establishing a dialog", and “thinking about the contents of the robot" was rated more positively than having a conversation with one robot.

### 2.2. Possibility of Emotion Amplification by Empathy between Humans and Robots

The feelings and impressions of users with the presence of agents were investigated. Matsumoto et al. [15] assessed the subjective evaluations of a participant watching a movie alongside an agent. The participants recorded more amplified emotions than after watching the movie by themselves. The movie was known to invoke certain emotions, which were reflected in the body movements and prosody information provided by the agent. It was suggested that the user’s emotion tends to become higher when watching the movie with the agent than when watching the movie alone. When the robot matches the human emotions invoked by the video, the resulting sense of empathy may improve the impression that a user feels not only for the robot but also for other phenomena.

### 2.3. Analysis of Comments on Social Media

Generally, when a TV program arouses viewers, the number of comments on SNS often increases rapidly. Accordingly, some researchers have estimated the emotional arousal level of TV programs from the number of comments on SNS [16,17,18]. Shamma et al. [19] proposed a method, in which it divides a TV-broadcasted debate into topic units and estimates the arousal points in each topic, detecting the climax point from the average and standard deviation of the number of comments in several sections before and after the peak point of transition of the number of SNS comments. However, it requires information on the number of comments for judging the peak points in future. Thus, the climax point cannot be detected in real time.

### 2.4. The Goal of the Present Study

We propose the method “emotion amplification by representing an emotive mood related to a TV scene” that is designed for TV chat agents aiming to improve the sociality of the agents proposed by Minami et al. [13], in which agents watch TV programs alongside with the users. For example, when watching live sports, such as a football game, the agent reacts positively to scores and fine play, creating a sense of affective empathy and presence with the user. As multiple agents improve the sense of dialog (Section 2.1), a lively mood is expected. We expect the agent continues short chats rather than continuing to talk about one topic with a user, and emotions and cheers can be mainly exchanged about common content.

We also propose an estimation model that can detect the climax point of TV in real time using the transitions in the number of comments posted on SNS. The agent can dynamically change its behavior to present the degree of excitement level to the user. The degree of excitement falls into four levels. These levels allow the agent to change its behavior during the climax in stages.

To evaluate our proposal, we asked participants to evaluate their impression of an agent implementing the proposed system and measured their subjective and objective (physiological) emotional responses while watching TV programs.

## 3. Proposed Method Implementation

From an emotional perspective, a high-positive and high-arousal state is the excited state while watching TV [20]. Valence (the qualitative component) and arousal (the intensity of the positive or negative emotion) represented the subjective emotional experience. Furthermore, when excitement is amplified, the arousal of positive emotional states is enhanced. Meanwhile, the excitement aroused by TV programs is assumed similar to the excitement aroused by SNS activity. To present the same excited mood as a user, we propose the following two requirements of an agent.

The agent estimates the degree of excitement elicited by a TV program in real time.The agent decides and operates its behavior from the estimated degree of excitement.

When multiple agents express an excited mood, the agents’ behavior must respond to the changing degree of the user’s excitement. In this study, the agents were four robots named RoBoHoN [21] (SHARP, Sakai, Japan). Each robot is a 19.5-cm-high easily carried robot, which is considered as a suitable daily-use agent. The agent utterances were created in the speech synthesis software VoiceText [22]. In this implementation, the agent neither recognizes the user’s natural language voice nor implements an automatic response.

### 3.1. Estimating the Degree of Excitement

As mentioned in Section 2.4, we propose an estimation model detecting the degree of excitement on TV in real time. By analyzing the number of SNS comments posted within a certain period, we estimated the degree of excitement. The comments sourced from Twitter [23] and Nico-nico Jikkyo (live Bulletin Board System in Japan) [24] are the comment source. Groups of 5 s are treated in sections because 5 s is the boundary frequency below which an application programming interface restricts access to SNS comments. Our estimation model inputs the number of SNS comments posted in 5 s intervals and then outputs the degree of excitement. The degree of excitement E(%) is a function of SNS comments $C_{i}$ in the last 5 s and the average number of comments μ during the last 30 min:$$E=\left\{\begin{array}{cc} \;\;\;\;1 & \left(4\mu \geq \frac{C_{i}-\mu }{3\mu }:more\;than\;max\;threshold\right)\\ \frac{C_{i}-\mu }{3\mu } & \left(other\right)\\ \;\;\;\;0 & \left(\frac{C_{i}-\mu }{3\mu }\leq \mu :less\;than\;min\;threshold\right) \end{array}\right.$$

Figure 1 shows the process flow of the model.

To elicit different behaviors in the agents, we divided the degrees of excitement into four levels. Figure 2 shows the output of the model and the criteria for classifying the levels.

### 3.2. Validation of the Estimation Model

To verify the accuracy of the created model, we compared its output with the correct degrees of the excitement elicited by TV programs, which were manually created in advance. For the accuracy verification, we used the TV content of the match between the National Football Team of Japan and Paraguay broadcasted on 5 September 2019. We selected 10 min from the first half (20 min) and 47 s of the relayed game in the same first half (30 min). The 10 min period, which includes two goals and one decisive chance, was considered as a suitable ground truth. The ground truth data were created by 10 male adults aged 23–30 years. In every second, the TV scene was judged by a binary value: 1 (exciting) or 0 (unexciting). However, in every interval (where one interval is equal to 5 s), the degree of excitement was scored on a scale from 0 to 50. Figure 3 compares the degrees of excitement estimated by our model and the ground truth.

As shown in Figure 4, the rising edge of the ground truth was detected 10 s earlier than the estimated degree of excitement. Two possible causes of this lag are detection of the degree of excitement after a delay of up to 5 s the time interval of one section and the difference in the evaluation timing between people and SNS users. The ground truth excitement evaluation was inputted manually, and the assessors tended to judge the excited mood when a ball pass would likely achieve a chance. However, many SNS users commented once a series of soccer games was finished. These different judgment tendencies in the manual and SNS data probably delayed the excitement degrees in the model from those of the human evaluation. However, provided that the degree of excitement is correctly estimated, the degrees of excitement in the TV program can be synchronized with the estimated excitement degrees by time-shifting the playback of the recorded program. Figure 4 compares the correct data and the data obtained after shifting the estimated degree of excitement by 10 s.

When the two datasets were synchronized, the timings of their rising edges were consistent at multiple locations. In a statistical investigation of the estimated versus ground truth data, the correlation coefficient was 0.714, confirming a strong correlation. Therefore, the proposed method can estimate the degree of excitement with a delay of 10 s.

### 3.3. Determining the Behavior of the Agents

There are two classifications of behavioral factors of TV chat agents: those related to speech and motion. Here, we focused on the factors related to speech. After analyzing how people gain excitement from TV programs, we determined the speech behavior parameters of TV chat agents. To analyze the excited moods of people, we video-recorded the behaviors of five male adults watching a soccer match, that is, the FIFA World Cup Japan vs Poland game held on 18 June 2018, for the first time. A snapshot of the video recording is presented in Figure 5.

We took the videos behind and in front (built-in camera of the laptop) of the participants and also used the build-in microphone of the laptop to record the speech of the participants. We then extracted the utterance frequency and prosody information (utterance interval, speech rate, volume, and pitch). These parameters are defined and calculated as follows:

**utterance interval:** length of the silent section between utterances by the same participant

**speech rate:** mean number of moras per second, obtained by dividing the number of mora by the utterance interval length

**volume:** mean value of the power spectrum of the voice section

**pitch:** The fundamental frequency ($f_{0}$) of the speech was extracted and averaged over the logarithmic domain. The average f0 after removing the unvoiced sections was determined as the pitch.

In Japan, mora means a “beat”, and each mora is temporally equivalent. It has the structure of “(consonant) and vowel”, but there are exceptions such as “n”, which has no vowels, and “tsu”, which is applied to the sound itself [25]. Through video analysis, we confirmed that the speech rate significantly depended on the strength of the positive emotion. In this study, we considered that the number of comments on SNS increased when the degree of excitement was raised in the TV program. Therefore, we focused only on the arousal dimension on the affect grid (Figure 6) [26] and ignored the strengths of pleasant and unpleasant emotions. Thus, the speech rate of the TV chat agent was fixed, and the VoiceText parameter was set to its default value (100). It was confirmed that the there were differences in utterance intervals, volumes, and pitches between the excited and non-excited moods. Thus, using the previous method [27], we converted volume and pitch to VoiceText-specific parameters for use with VoiceText. Table 1 listed the behavior parameters obtained by the analysis. To represent levels 3 and 0, we computed the mean values of the participants in the excited and non-excited moods, respectively. The parameters at levels 1 and 2 were obtained by linear interpolation between those of levels 0 and 3. The agents dynamically change their parameters according to the estimated degree of excitement, and their speech reflects the excitement level of the user.

Since the comment of the robot is randomly selected from a group of comments made by SNS users who watched the TV content, the mainstream comments are generally selected according to the TV content at that time. In this experiment, we selected SNS sources that are used by users whose favorite teams are the same as the robot users, so the comments of SNS users who watched the TV content will have the same tendency as the users. Therefore, even if we randomly select a speech comment from that group of comments, a comment that is generally appropriate for TV content will be selected.

## 4. Experimental Conditions

To verify the amplification of emotions and motivation of human–agent interactions of our TV chat agents, we evaluated the emotional responses of a user while watching TV programs with the agents. Specifically, using the football game contents, we evaluated the subjective and objective responses. Using the affect grid, the subjective valence, and the arousal (which reflects the energy level of the emotion [26]), we qualitatively assessed the subjective emotion responses. To complement the subjective rating, which can be biased by factors such as the demand characteristics [28,29], we measured some physiological responses related to emotion, namely, the electromyography (EMG) signals from the corrugator supercilii (i.e., brow lowering) and zygomatic major (i.e., lip-corner pulling) muscles, and the skin conductance level (SCL). The muscle signals and SCL reflect the emotional valence and arousal responses, respectively. Psychophysiological evidence indicates that relative to emotionally neutral events, positively arousing events reduce the activity of brow lowering, increase the activity of lip-corner pulling, and raise the SCL [20,30].

### 4.1. Participants

One of the factors that may affect the impression evaluation is the participants’ interest in a TV program. Thus, we only recruited participants who met the experimental criterion: “being interested in soccer”. A total of 24 Japanese volunteers (4 women, 19 men, and 1 unknown gender; all aged in their twenties) was recruited for the study. We determined the required sample size in an a priori power analysis using G*Power software ver. 3.1.9.2 [31] based on a preliminary experiment, in which a similar procedure and facial EMG recordings were implemented on a different sample (*n* = 5). From the results, the effect size was estimated at an α level of 0.05 with power (1−β) of 0.80 was assumed. The power analysis revealed that 24 participants were needed for a meaningful study. This study was conducted under the Declaration of Helsinki and was approved by the Ethics Committee of the Nara Institute of Science and Technology. Written informed consent was obtained from all participants, and all participants received a detailed explanation of the experimental procedure.

The objective EMG data were collected using sets of pre-gelled, self-adhesive 0.25-cm Ag/AgCl electrodes (Prokidai, Soraku-gun, Japan) and an EMG-025 amplifier (Harada Electronic Industry, Sapporo, Japan). The SCL was recorded using pre-gelled, self-adhesive 1.0 cm Ag/AgCl electrodes (Vitrode F, Nihonkoden, Tokyo, Japan) and a Model 2701 BioDerm Skin Conductance Meter (UFI, Morro Bay, CA, USA). Finally, unobtrusive video recording was performed by a digital web camera (HD1080P, Logicool, Tokyo, Japan). The data were sampled by a PowerLab 16/35 data acquisition system and LabChart Pro v8.0 software (AD Instruments, Dunedin, New Zealand).

### 4.2. Agent Conditions

The experimental results were acquired and compared under the following three conditions:

**No agent:** the participants watched the video alone.

**Neutral agents:** the participants watched the video with four agents speaking with mixed behavior parameters.

**Excited agents (proposed method):** the participants watched the video with four agents responded to an exciting scene in the video, enhancing the participants’ excitement level.

Under the neutral agent condition, the speech rate, pitch, and volume of the agents’ utterance were fixed at 100 in VoiceText.

It is expected that the arousal level and user’s motivation to use the agents to be highest during interactions with the excited agents, lower with the neutral agents, and lowest with no agent. The agents recited the SNS comments with the utterance parameters explained in Section 3.3, which depended on the estimated degree of excitement. The agents then uttered the voice generated in advance based on the degree of excitement. As the agent was operated in real time, the timing of our estimation model was shifted by 10 s.

### 4.3. TV Content

To convey the exciting scenes to the participants, The titles of the selected games are as follows.

Japan versus Paraguay (broadcasted on 5 September 2019)Japan versus Mongolia (broadcasted on 10 October 2019)Japan versus Kyrgyzstan (broadcasted on 14 November 2019)Japan versus China (broadcasted on 10 December 2019)

These contents included exciting scenes at well-clarified timings, such as chance scenes, fine plays, and goal scenes, which are expected to increase arousal and pleasant emotions on the affect grid. We applied the proposed model that estimates the degree of excitement to the SNS comments related to the game broadcasts and then selected the scenes determined to elicit an excited mood. We prepared 1.5- to 2-min 12 videos that captured these exciting scenes. Figure 7 shows the examples of the agent’s utterance for each level of excitement and the expected interaction with a participant.

### 4.4. Procedure

The experiment was conducted individually in a soundproof room (Science Cabin, Takahashi Kensetsu, Tokyo, Japan), where each participant wore a physiological measuring device while watching the video (Figure 8).

Three trials (one under each condition) were performed in each session, and each participant underwent four sessions. We randomized the order of TV programs and conditions among the participants. After each trial, participants rated their emotional experiences on a 9×9 affect grid. Using the seven-point Likert scale (1: not at all, 7: very much), we also evaluated the agent by answering the following two questions: “Were the agents in an excited mood?” and “Did you feel empathy with the agents?” Thus, for subjectively evaluating the user’s impression of the agents were created by a proprietary methodology. The aim was to reduce the work burden on the participant by acquiring the minimum required information for evaluating the agents of the experiment. To support the subjective results obtained in the experiment, we acquired the biological information of the participants—skin conductance and facial EMG—as objective indexes. After each session, the participants also selected one among the three conditions that they would care to repeat. Finally, the participants were asked to briefly and freely describe their responses during the session.

### 4.5. Analysis

In every participant, the mean ratings and selection percentages were calculated under each condition. The physiological data were sampled for 10 s during the pre-stimulus baseline period, which is immediately before the presentation and duration of the stimulus period. We excluded the data from one participant because of equipment error. After rectifying the EMG data, the SCL and EMG data of each trial were baseline-corrected to their average values over the pre-stimulus period, and then averaged. The data outside >|3|SD in each participant were removed as outliers.

Our hypotheses were tested by independent *t*-tests (one-tailed), in which we detected differences between the no agent and neutral agent conditions, and between the no agent and excited agent conditions. We also exploratorily compared other directions and conditions using dependent *t*-tests (two-tailed). All results were considered significant at p< 0.05.

## 5. Results

### 5.1. Emotional Ratings

Among the subjective ratings of valence and arousal while watching the TV contents (Figure 9), only the arousal ratings were significantly different between the no-agent and excited agent conditions (*t*(23) = 2.99, p< 0.005).

### 5.2. Physiological Activity

Among the physiological activity ratings, while watching the TV contents (Figure 10), only the SCL responses were significantly different between the no-agent and excited agent conditions (*t*(22) = 1.65, p< 0.05).

### 5.3. Evaluation of Agents

Figure 11 shows the excitement and empathy degrees rated by the participants interacting with the neutral and excited agents. Only the excitement ratings significantly differed between the two conditions (*t*(23) = 4.84, p< 0.001), indicating that excited agents were perceived as more excited than neutral agents.

### 5.4. Motivation to Interact with Agents

Figure 12 shows the percentages of the conditions selected for reuse by the participants at the end of each session. There were significant differences between the no-agent and the proposed method (*t*(23) = 2.98, p< 0.005) and between the no-agent and neutral conditions (*t*(23) = 2.26, p< 0.05). These results indicate that watching TV contents alongside both the proposed and neutral agents motivated further use of the TV chat agent.

## 6. Discussion

### 6.1. Enhanced Emotional Arousal Watching TV Alongside Excited Agents

Relative to the no agent case, the subjective ratings (Figure 9) and physiological measures (Figure 10) demonstrated that the excited agents significantly enhanced the subjective and physiological emotional signals of the participants. However, the valence measures were not significantly affected by the agents, indicating that the agents presenting an excited mood can amplify the arousal level of users without qualitatively changing their emotional experience. These results are consistent with previous findings, which reported that dialog agents could improve the positive experience of people watching TV or movies [14,15]. However, these studies did not manipulate the shared excitement levels in the agents nor assess the physiological aspects of the emotional responses, which likely introduce less bias than subjective ratings [28,29]. To our knowledge, we present the first evidence that agents sharing human excitement can amplify users’ subjective and physiological emotional arousal while a user is watching TV.

### 6.2. Enhanced Excitement Recognition in Excited Agents

According to the impression ratings (Figure 11), participants interacting with the excited agents became significantly more excited than participants interacting with the neutral agents. Therefore, our proposed agents conveyed excitement in their prosody. In contrast to our expectation, affective empathy was not significantly enhanced by the excited agents. To interpret this result, we reverted to the free responses of the participants. Some participants mentioned that the agents’ comments were generally disappointing and discouraging. Based on these opinions, we speculated that SNS-based comments might not effectively induce empathy in general.

### 6.3. Improving Motivational Use of the Excited Agents

Figure 12, shows that the participants were significantly more motivated to reuse the excited agent than the neutral agent, or to re-watch the video alone. It was considered that the emotional evaluation in the proposed system significantly enhanced the arousal level of participants, and that the dynamic behavior of the agents gave the impression of excited mood. Consistent with this interpretation, some participants mentioned that when watching TV without the agents, they missed the excited agents. Collectively, these data support our proposal, that agents presenting an excited mood can inspire the reuse of those agents.

### 6.4. Limitation

A limitation of this study should be acknowledged. The result of this study is that it was obtained under the condition that the delay of 10 s was not considered. This delay is possible to affect the user’s impression of the robot, so a method that resolves the delay may be needed. To remove the delay, we could control the excitement timing of the agents through the time-shift playback function of the TV program. We can play back the recorded video while recording a TV program by the time-shift playback. Alternatively, we could give (in advance) the agent a fixed phrase appropriate for the situation, which could be spoken when the situation was detected.

## 7. Conclusions

We proposed TV chat agents that learn an excited mood based on SNS comments related to TV programs, and present the same mood to users watching the same TV contents. The aim was to improve the user’s motivation to interact with the agents. We also proposed that the degree of excitement is determined from the number of SNS comments per second, and is classified into one of the four levels. Based on the estimated degree of excitement, the agent dynamically changes the interval, pitch, and volume of its utterances and presents an excited or non-excited mood to users. To verify the effectiveness of the excited agents, we compared the reactions of participants under the excited agent, no agent, and neutral agent conditions. We performed subjective evaluations using affect grid and a questionnaire and objective evaluations using physiological responses.

The arousal level becomes significantly higher in participants interacting with the excited agent than under the other experimental conditions; however, participants’ emotional experiences were not significantly different among the conditions.

The participants also expressed more desire to reuse the excited agents than to reuse the neutral agents or re-watch the TV contents alone. Providing a lively mood with no delay might further amplify the user’s emotions and enhance their motivation to interact with the agent.

During the present experiment, the participants watched a video recording of a soccer game targeting the arousal-pleasure dimensions of the affect grid. The mood presentation target of the TV chat agents in this experiment was excitement. In future work, we must comprehensively investigate whether the agent can enhance the user’s emotions while watching other TV programs such as variety programs and news. This exploration would confirm the generality of the agent to any TV program. Another perspective is the impression of moods other than excitement; for example, we could evaluate whether the agents can enhance laughter. SNS comments include language and slang representative of laughter, which can be inputted to the model. Like excitement, the presentation of laughter might increase the user’s emotions and motivation to interact with the agent. In this way, the moods of TV programs presented by TV chat agents can be comprehensively evaluated. At this time, the content uttered by the agents is randomly selected, so comments that are favorable to the user should be filtered to enhance the user’s impression of the agent by the latest SNS analysis.

## Figures and Tables

**Figure 1 sensors-20-07330-f001:**
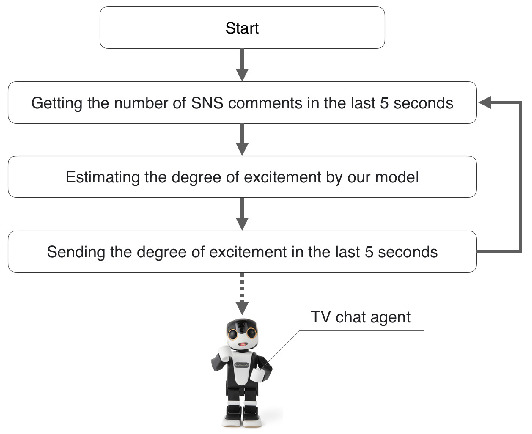
The processing flow of our model.

**Figure 2 sensors-20-07330-f002:**
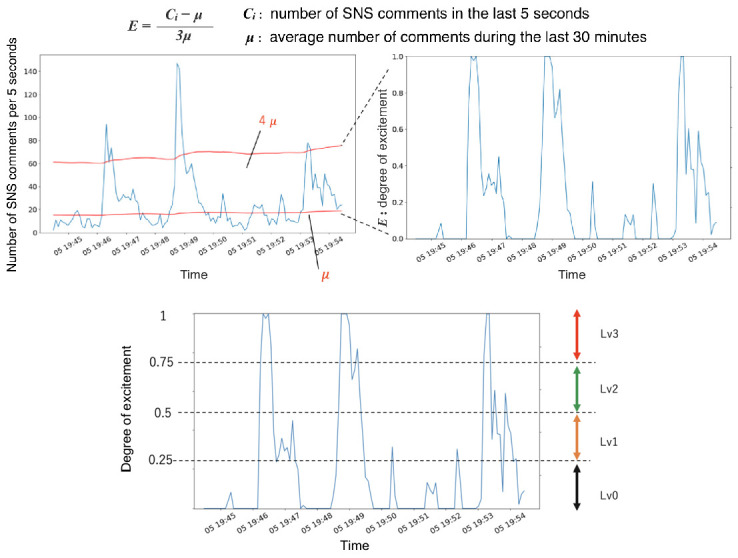
An example of a transition in the number of SNS comments and output of our model (upper image), and excitement variations during a television program sorted into levels (lower image).

**Figure 3 sensors-20-07330-f003:**
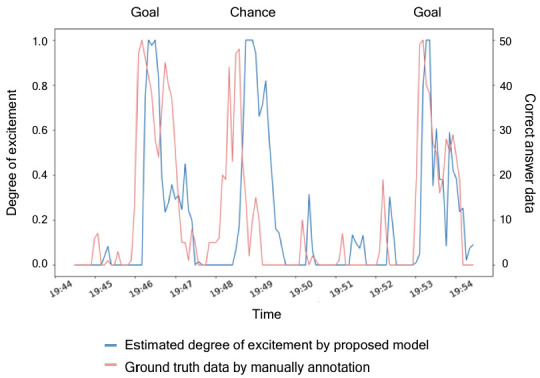
Estimated degree of excitement (blue lines) and the ground truth (pink lines).

**Figure 4 sensors-20-07330-f004:**
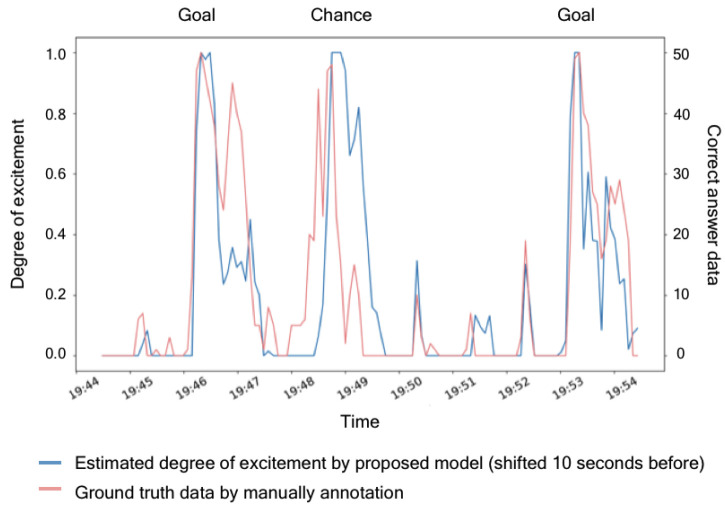
Estimated degree of excitement (blue lines) and the ground truth (pink lines) after a 10 s shift.

**Figure 5 sensors-20-07330-f005:**
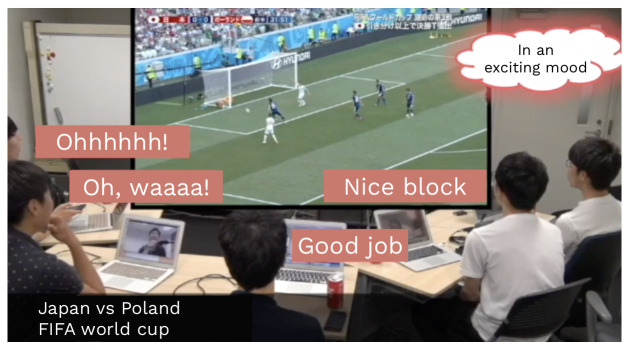
Snapshot of the video recording of excitement levels by participants watching a soccer match.

**Figure 6 sensors-20-07330-f006:**
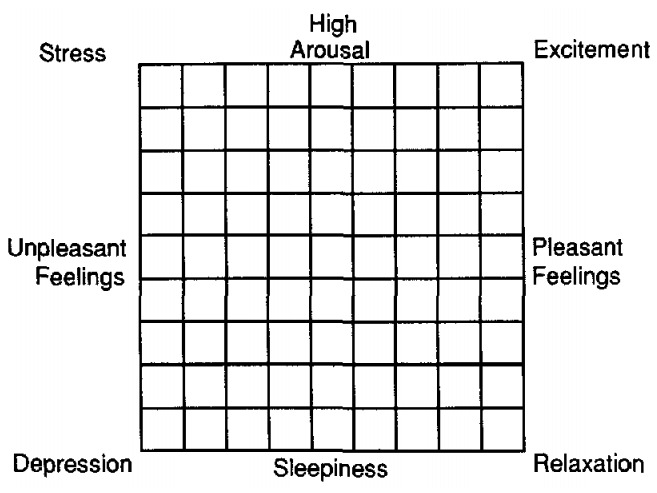
Affect grid for quantifying emotions.

**Figure 7 sensors-20-07330-f007:**
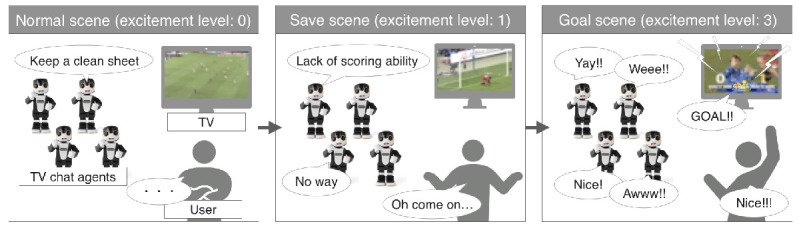
Examples of interaction between the TV chat agents and a user.

**Figure 8 sensors-20-07330-f008:**
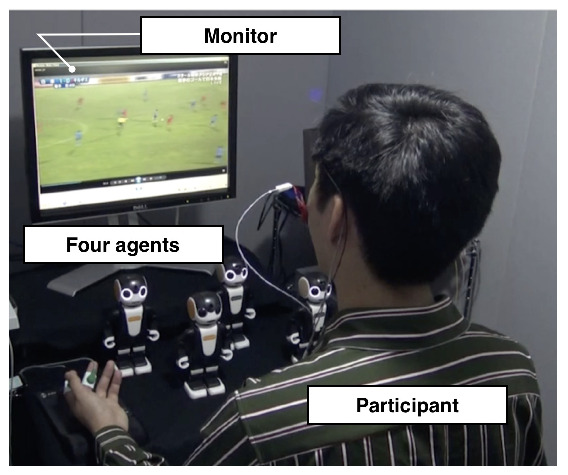
An overview of the soundproof room.

**Figure 9 sensors-20-07330-f009:**
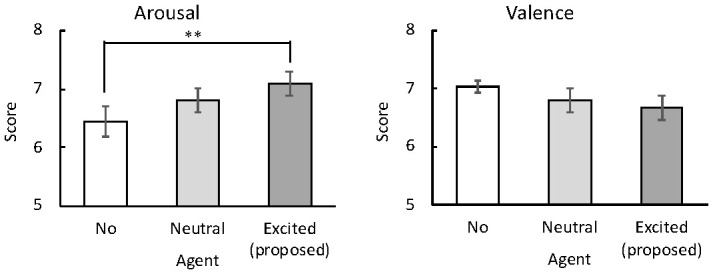
Mean subjective ratings (with standard error bars) of excitement (left image) and empathy (right image) for the agents by the participants, rated on the affect grid. ** p< 0.005.

**Figure 10 sensors-20-07330-f010:**
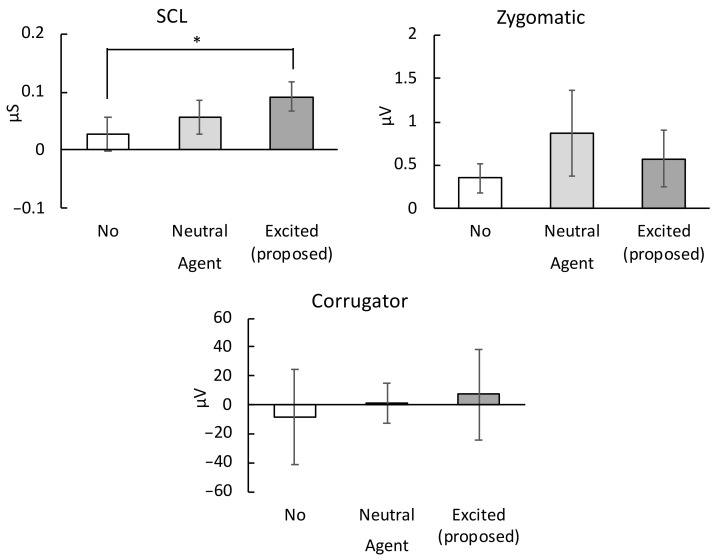
Mean physiological responses (with standard error bars) measured while watching the TV contents: electromyography data of the skin conductance level (SCL) (upper left image), zygomatic major (upper right image), and corrugator supercilii muscle (lower image). * p< 0.05.

**Figure 11 sensors-20-07330-f011:**
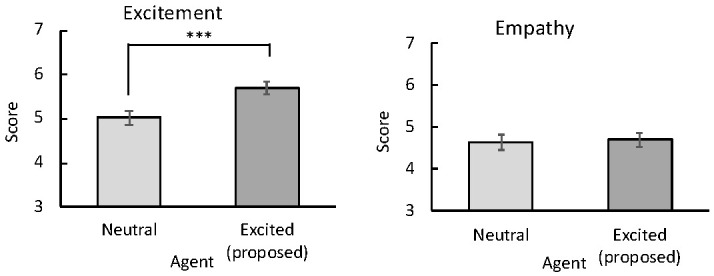
Mean (with standard error) subjective ratings of excitement (left image) and empathy (right image) for agents. *** p< 0.001.

**Figure 12 sensors-20-07330-f012:**
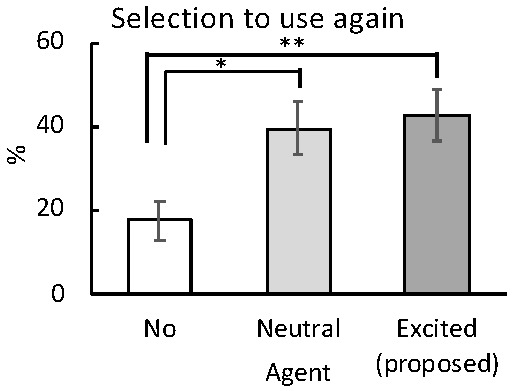
Participants’ mean re-selection percentages (with standard error bars) under each experimental condition. ** p< 0.01; * p< 0.05.

**Table 1 sensors-20-07330-t001:** This is a table caption. Tables should be placed in the main text near to the first time they are cited.

	Utterance Interval (s)	Pitch	Volume	Speech Rate
Level 3: the parameters when the participants were excited	4	123	132	100
Level 2	7	117	118	
Level 1	10	111	105	
Level 0: the parameters when the participants were not excited	13	105	92	

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
