# Peer review of "Positive Emotion Amplification by Representing Excitement Scene with TV Chat Agents"

_sensors, 2020, doi:10.3390/s20247330_

Round 1

Reviewer 1 Report

This paper proposed a method for increasing excited states in emotional experiences by TV watchers through robots using speech synthesis. Results were evaluated by comparing the correct degrees of excitement elicited by TV programs manually created in advance. Results showed more arousal when using the proposed method.

Please check the use of "make a user more exciting" in the abstract.

In abstract: "make a user more exciting" should be "more excited". Or perhaps, it refers to "making the experience more exciting to the user".
In page 2, line 73: "Other researches". Although not incorrect, the use in plural of "research" is uncommon.
page 3, line 81: "the impression of the robot" gives the idea that the robot has an impression, where I assume the text refers to the robot gives an impression. Please check this part.
84: "explosively" is vague. Please select a better description.
109: "We further assume *that* when excitement..."
119: Reference is missing: "As mentioned in ... we propose"
In page 4, formula for E is not clearly defined. Please add the corresponding words: "if... if... else". Please add afterwards an example of the average number of comments.

161: As the concept of mora is more likely to be used for Japanese, please provide a reference for its use, so that readers can understand it properly.

Caption of Figure 7 says: "ratedon"
294: additional space before comma

This is a very interesting work that would be greatly benefited from adding an URL where videos can be seen. It is really important to appreciate the arousal and excitement of agents, as well as human participants.

Author Response

Thank you for your comments dated November 10, 2020, regarding our manuscript (sensors-1002170).

We thank you for their helpful comments.

We have revised our manuscript accordingly.

Reviewer 2 Report

The manuscript discusses an excitement arousal system for TV sports audience through their interaction with robotic agents. Overall the presented work is well defined and sound.

A general comment about the text is that the authors should provide appropriate references for any statements they make. For example, the statement “many researchers have proposed daily communication activation by agents such as dialog robots” should be accompanied by the appropriate reference, otherwise it can only be perceived as generic having no impact in respect of their argumentation.

The related work provided by the authors needs to be improved, especially regarding the sub-section 2.3 ‘Analysis of comments on social media’.

Specifically, although a paramount related work has been conducted in the field the authors provide selectively a very limited number of references, excluding research works which constitute state of the art in the field.

For the description of the proposed system, the authors are strongly advised to provide more details regarding the interaction supported between the users and the robots, clarifying whether the system employs natural language dialogues, further discussing the algorithm used for generating robots’ responses as well as how the response information that is related to the topics of discussion is constructed.

It would be very beneficial for the readers if the authors could give an example of the interaction provided.

Furthermore, the authors should further elaborate on their decision to base their analysis for extracting the excitement periods from the relative SNS only on their number, without taking into account qualitative results that could be extracted from the SNS’s sentiment analysis.

In this respect, the authors should clarify how the proposed system avoids potential false positive results, i.e. how it can filter out a big number of posts that might be sent, massively at a specific point of time, by disappointed fans declaring their disappointment for a lost chance of their team, or a contentious decision against their team.

Additionally, the authors should further argue on how the proposed system addresses the fact that a 10 second delay exists between the actual excitement arousal and the estimated, which can lead to awkward delayed responses of the robots.

Related to the qualitative evaluation the authors should provide further details on their decision to follow a proprietary methodology and not a well-established one, such as SUS. Finally, a proof reading is needed, since there are some typos:

  • line 27: “heir” instead of “their”
  • line 119: missing word in the sentence starting “As mentioned in “, just after the sub-section title

Author Response

Thank you for your comments dated November 23, 2020, regarding our manuscript (sensors-1002170).

We thank you for their helpful comments.

We have revised our manuscript accordingly.

Round 2

Reviewer 2 Report

The revised version is indeed improved in terms of clarity and comprehension for the reader. However, the authors couldn't explain my question regarding how the system detects which is the appropriate pool of messages to select so that the chats uttered by the agents will fit in the moment. For example in Figure 7, for the excitement level 1, the agents utter context-related messages that fit in the moment (save scene) and not a random picked message as the authors claim. In order for the system to provide this functionality it needs to understand what is discussed about in the SNS posted at the time that the scene happens. Thus, this functionality requires a content-analysis approach and cannot be addressed only based on the statistical analysis of the SNS number. 

Author Response

Thank you again for your valuable feedback. I am very sorry for not being able to give you a sufficient answer.

Round 3

Reviewer 2 Report

I'm happy to confirm that the new version of the manuscript includes the necessary clarifications with regard to my comments, thus the manuscript in my opinion is ready for publication.